# Predicting art university students' entrepreneurial intention: A hybrid SEM–ANN approach

Yiliang Cao[1], Jie Zhang [ID][2]*

1 School of Art & Design, Liaoning Economic Management Cadre College, Shenyang, China,
2 Department of Human Resource, Guangdong University of Petrochemical Technology, Maoming, China

* ming347817@163.com

## Abstract

In recent years, academics and policymakers have increasingly focused on entrepreneurial behavior among university students. While existing studies have explored the entrepreneurial intention (EI) of students from various academic disciplines, few have specifically examined the EI of art university students. Based on the Diffusion of Innovations Theory (DOI) and the Theory of Planned Behavior (TPB), this study explores the factors influencing art university students' EI and assesses each factor's relative importance. This study employed a structural questionnaire to survey 273 students from three universities in Liaoning Province, China, measuring eight constructs: relative advantage (RA), observability (OB), compatibility (CO), entrepreneurial motivation (EM), entrepreneurial attitude (EA), subjective norms (SN), perceived behavioral control (PBC), and EI. Data analysis was conducted using Structural Equation Modeling (SEM) and Artificial Neural Networks (ANN). The results show that, among the direct significant predictors of EI, PBC has the strongest influence, followed by EA, SN, and EM. Additionally, all predictive constructs accounted for 60% of the variance in the EI of art university students. The ANN analysis revealed the following normalized importance ranking of all predictive constructs: PBC (100%), EA (70.8%), SN (57.6%), RA (43.1%), and EM (31.2%). This study not only fills the gap in research on the EI of art university students but also provides valuable insights for developing targeted strategies to foster entrepreneurship among this group.

## 1. Introduction

Global economic and technological advancements have made entrepreneurship a key driver of societal and economic development [1,2]. According to the Global Entrepreneurship Monitor 2023 report, the global entrepreneurial activity index has increased by nearly 15% over the past three years, particularly in high-tech and innovative sectors [1,2]. University students, with their higher education and innovative capabilities, possess unique advantages in entrepreneurship, making their

**Data availability statement:** All relevant data are within the paper and Supporting Information files.

**Funding:** The author(s) received no specific funding for this work.

**Competing interests:** The authors have declared that no competing interests exist.

entrepreneurial activities an increasing focus of academic research and policy development [3]. For instance, a study by Deng and Wang [4] found that university-educated entrepreneurs have a survival rate of 78% in the early stages of their ventures, significantly higher than that of other groups. However, the factors influencing university students' entrepreneurial intention (EI) are complex and diverse. A deeper understanding of these factors is crucial for stimulating entrepreneurial enthusiasm among students and improving the success rate of their ventures [5,6].

In recent years, researchers have focused on the EI of university students from various academic disciplines, including engineering [7], business [8], sports [9], medicine [10], and female students [11,12]. Furthermore, scholars have employed a variety of theoretical frameworks to explain and predict EI, such as the Theory of Planned Behavior (TPB) [13], Social Cognitive Theory [14], the Technology Acceptance Model (TAM) [15], the Unified Theory of Acceptance and Use of Technology (UTAUT) [16], Diffusion of Innovation Theory(DOI) [17], Entrepreneurial Education Theory [18], and Self-Determination Theory [19]. In terms of data processing techniques, the main methods used to study university students' EI include Structural Equation Modeling (SEM) [20], Multilevel Regression Analysis [21], Path Analysis [22], Artificial Neural Networks (ANN) [23], and Cluster Analysis [24].

Although existing research has provided abundant theoretical and empirical support for understanding university students' EI, studies focusing on art university students as a unique group still require further exploration to fill the gap in current research. First, while a considerable amount of research has examined the EI of students from various academic disciplines, there remains a clear lack of studies specifically addressing the EI of art university students. Art students typically possess strong creativity and unique ways of thinking, with their entrepreneurial pursuits often diverging from traditional industries. They are more likely to focus on cultural and creative industries, art design, and performing arts. These fields play a significant role in driving social innovation and the development of the cultural industry. Second, there is limited research that integrates the DOI and the TPB to explore the EI of art university students, resulting in a less comprehensive understanding of this group's entrepreneurial motivation (EM). Integrating these two theories allows for a more holistic understanding of how art students acquire creative information and develop unique EM during the entrepreneurial process, thereby enhancing the depth and accuracy of the research. This approach enables the design of information dissemination strategies to be more aligned with the characteristics of the creative industries. It strengthens their confidence and capabilities in arts entrepreneurship by fully considering the challenges related to creative expression and market demand [25]. Finally, existing research predominantly relies on traditional statistical methods, such as SEM, which struggle to uncover complex non-linear relationships, thus limiting the predictive accuracy of EI. Integrating SEM with ANN can address this limitation. SEM effectively identifies and validates linear relationships and hypotheses between variables, while ANN excels at handling non-linear and complex relationships. Combining these two data processing techniques allows a more comprehensive and accurate analysis of art university students' EI, providing more insightful research outcomes [26].

Based on the integrated DOI and TPB, this study employs SEM and ANN to analyze self-reported data on the EI of art university students. In alignment with the research objectives, the following research questions are proposed:

(1) What factors significantly influence the EI of art university students?

(2) To what extent do these factors explain the EI of art university students?

(3) What is the standardized importance of each factor in predicting the EI of art university students?

Compared to existing research, the main contributions of this study are as follows. First, it fills the theoretical gap in research on the EI of art university students. This study focuses on the unique entrepreneurial behaviors of art students, addressing the gap in the existing literature regarding this group and providing important theoretical guidance for entrepreneurship education in art institutions. Second, it integrates the DOI and the TPB to construct a novel theoretical framework. By combining these two theories, this study introduces a new theoretical framework for understanding the EI of art university students, offering a more comprehensive analysis of the interaction between the dissemination of innovation information and individual behavioral intentions in the face of entrepreneurial opportunities. This further enriches theoretical research in this field. Third, integrating SEM and ANN provides robust theoretical evidence. This study accurately identifies the key factors influencing the EI of art university students through the fusion of these techniques. It offers more robust data support and theoretical foundations for future research and practical applications.

The structure of this paper is as follows. The second section presents the literature review and research hypotheses. The third section describes the research methodology, including the sample, measurements, and data analysis. The fourth section introduces the results of this study. The fifth section discusses the research findings, including an analysis of the results, limitations, and suggestions for future research directions. Finally, the conclusion of the study is presented.

## 2. Literature review and research hypotheses

### 2.1. Diffusion of innovation theory

DOI, proposed by Rogers [27], aims to explain how ideas or products gain momentum and spread within a specific population or social system over time. This theory contains five key dimensions: relative advantage (RA), compatibility (CO), complexity, trialability, and observability(OB) (Fig 1). RA refers to the degree to which an innovation is perceived as better than existing ideas, practices, or products. CO indicates how an innovation aligns with potential adopters' values, experiences, and needs. Complexity refers to the difficulty of understanding or using the innovation. Trialability is the extent to which an innovation can be tested or experimented before full adoption. OB refers to the degree to which the results or benefits of an innovation are visible to others.

In previous studies, DOI has been widely applied in various contexts to examine users' behavioral intentions, including virtual reality [28], knowledge management systems [29], open educational resources [30], digital entrepreneurship [31], autonomous vehicles [32], and internet usage [33]. For example, based on the DOI, Modgil, Dwivedi [31] conducted semi-structured interviews to analyze the emerging field of digital entrepreneurship. The results indicated that digital entrepreneurship opportunities are gradually emerging in four sectors: technology, healthcare, entertainment, and e-commerce. In addition to the core dimensions of DOI, researchers have incorporated other constructs into the model to enhance its predictive power. These include perceived attitude, subjective norms (SN) [28], knowledge or information quality [29], risk of sustainability, risk of reputation, status symbol [34], perceived usefulness, visibility [32], well-being, and perceived value [33]. Furthermore, DOI can be integrated with other models to explain user behavioral intentions, such as the TPB [28], TAM [32], the Uses and Gratifications Theory [35], and UTAUT2 [36]. For example, based on an integrated model of DOI and the TAM, Yuen, Cai [32] examined the factors influencing users' intention to use autonomous vehicles. The results indicated that all predictive constructs accounted for 75% of the variance in behavioral intentions.

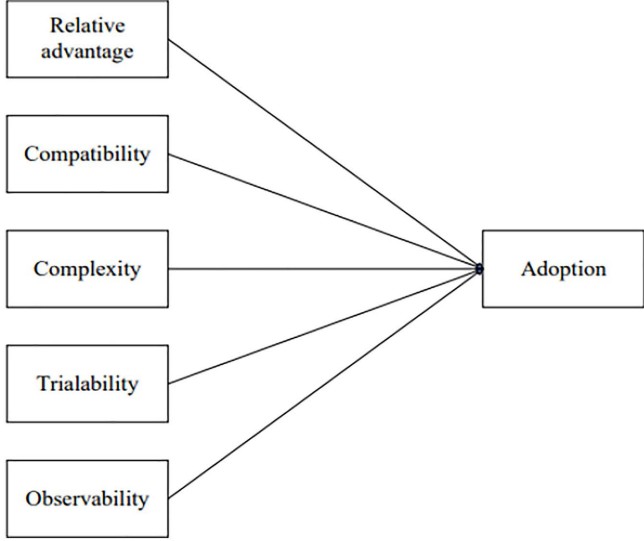

**Fig 1. DOI model [27].**

## 2.2 Theory of planned behavior

The TPB, proposed by Ajzen [37], aims to explain and predict individuals' intentions and actual behaviors regarding specific actions. The TPB has five key constructs: attitude, SN, perceived behavioral control (PBC), behavioral intention, and actual use (Fig 2). According to Ajzen [37], attitude refers to an individual's positive or negative evaluation of performing a behavior. SN denotes the perceived social pressure to perform or not perform a behavior. PBC refers to the perceived ease or difficulty of performing a behavior, reflecting an individual's perception of control over the behavior. Behavioral intention represents the individual's intention or determination to perform a specific behavior. Actual use refers to the actual performance of the behavior or action.

In previous studies, the TPB has been widely applied in various contexts to examine users' behavioral intentions, including EI [38], entrepreneurial education [39], family pressure [40], and mobile medical platforms [41]. Mensah, Khan [38] explored the factors influencing university students' EI and the moderating effect of motivations between these factors and EI. In addition to the core constructs of the TPB, researchers have incorporated various factors into the TPB to enhance its predictive power. These include entrepreneurship education, entrepreneurial self-efficacy [38], entrepreneurial passion, creativity [39], institutional environment [40], entrepreneurial competence [42], behavioral beliefs, normative beliefs [43], attitude, social influence, and perceived privacy risks [41]. Furthermore, the TPB can be integrated with other models to explain users' behavioral intentions, such as Flow Theory [44], TAM [41], Social Cognitive Theory [45], and Protection Motivation Theory [46]. Based on TPB and Flow Theory, Wu and Tien [44] revealed the mediating effect of learners' flow experiences and attitudes on the intention-behavior relationship, providing a deeper understanding of how entrepreneurship education influences students' exploratory entrepreneurial behavior.

Studies also show that TPB constructs vary across disciplines: business students often have higher PBC due to formal entrepreneurship training, while students in medicine or law may show weaker entrepreneurial attitudes due to fixed career paths [47]. In contrast, art students are usually driven by intrinsic creativity and personal expression, which can strengthen their entrepreneurial attitude (EA) but also make them more sensitive to social expectations. Family and institutional influences may significantly shape SN in collectivist societies like China. This study builds on these insights by exploring how TPB operates among Chinese art university students. In addition to widely cited TPB-based models,

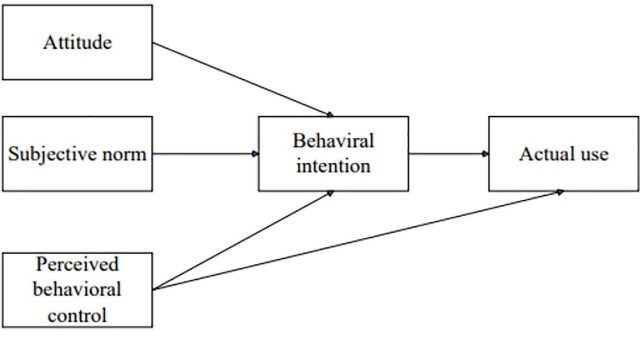

**Fig 2. TPB model [37].**

several comparative studies highlight the importance of national and cultural contexts in shaping EI. For example, Pejic Bach, Aleksic [48] examined university students in Slovenia and found that innovative cognitive style moderated the relationships between TPB constructs and EI. This study underscored the influence of individual thinking patterns in a developed European context. Similarly, Hossain, Tabash [49] investigated EI among Gen Z students in Bangladesh and emphasized the constraining role of economic and institutional barriers. These findings contrast with studies from more developed economies, illustrating the value of context-sensitive entrepreneurship research. By incorporating such comparative perspectives, our study gains broader relevance, particularly in understanding how TPB-based models operate in the Chinese context of art university students.

Integrating the DOI with the TPB provides a more comprehensive theoretical framework for studying the EI of art students. The DOI emphasizes how innovations spread within social systems, revealing how individuals' acceptance of new ideas influences their behavioral decisions, particularly when understanding students' attitudes and behaviors in the face of emerging entrepreneurial opportunities [50]. On the other hand, TPB focuses on forming individual behavioral intentions, emphasizing the roles of attitude, SN, and PBC in entrepreneurial decision-making [37]. When these two theories are combined, the DOI offers a perspective on external environmental factors influencing EI, especially in creative industries, where emerging business models and technologies may significantly impact students' entrepreneurial aspirations. Meanwhile, TPB further explains individuals' internal motivations and PBC in specific contexts. Integrating these two frameworks helps reveal the multidimensional factors that shape EI among art students in complex social, economic, and cultural environments. This, in turn, provides theoretical support for fostering entrepreneurial awareness and advancing entrepreneurship education [51].

Although the DOI theory and the TPB differ in their research foci—with the former emphasizing adoption behavior and the latter focusing on behavioral intention—existing studies have pointed out that in entrepreneurship, actual adoption behavior is often difficult to observe directly at the student stage. As such, EI is a valid proxy indicator for the likelihood of adoption [37,50]. Consequently, many researchers have adopted key predictors from the DOI framework (e.g., RA, CO, OB) as antecedents of behavioral intention rather than adoption behavior itself [28,31], thereby achieving a theoretical transformation and integration. Following this approach, the present study treats the core innovation characteristics from DOI (i.e., RA, CO, and OB) as antecedents of EA, which in turn predicts EI through the TPB framework. This is because, during the entrepreneurial cognition process, students' perceptions of opportunity characteristics—such as whether the innovation offers advantages or is easily observable—influence their attitudinal evaluations, which are then translated into intention. Therefore, this study links DOI and TPB at the attitudinal rather than behavioral level, aiming to capture the indirect pathway through which innovation characteristics shape EI.

## 2.3. Hypotheses

In this study, CO refers to the degree of alignment between entrepreneurial ideas or opportunities and university students' interests and career goals. According to the DOI, students are more likely to accept entrepreneurial opportunities with higher CO, leading to more positive EA [50]. Moreover, previous studies have demonstrated the relationship between CO and EA [52,53]. Ezeh, Nkamnebe [53] explored how the CO of innovations influences university students' EA. The results showed that students exhibit more positive EA when innovations align well with personal values and career goals. Based on this, this study hypothesizes that the higher the degree of alignment between entrepreneurial ideas or opportunities and the interests and goals of art students, the more positive their EA will be. Therefore, the following hypothesis is proposed:

**H1:** CO significantly positively influences the EA of art university students.

In this study, RA refers to the economic rewards, social prestige, or personal growth benefits that entrepreneurship can bring. According to the DOI, when university students perceive entrepreneurship as offering higher economic returns or greater personal development opportunities, their EA will become more positive [50]. Previous research has shown that RA significantly predicts EA [54,55]. Ramsey, Rutti [55] suggested that students' EA is significantly enhanced when entrepreneurial opportunities are perceived as more competitive than traditional career paths. Based on this, this study hypothesizes that the higher the RA of entrepreneurship, the more positive the EA of art university students will be. Therefore, the following hypothesis is proposed:

**H2:** RA significantly positively influences the EA of art university students.

In this study, OB refers to the widespread dissemination of successful entrepreneurial cases or the direct benefits of entrepreneurial activities. According to the DOI, the OB of successful entrepreneurial cases enhances university students' confidence and interest in entrepreneurship, fostering a positive EA [50]. Moreover, previous research has shown that OB can effectively enhance EA [56,57]. Bae, Lee [57] examined the factors influencing EI by combining descriptive norms and expected inaction. Their findings revealed that the OB of entrepreneurship directly impacts EA and strengthens EI through SN. Based on this, this study hypothesizes that the higher the OB of entrepreneurship, the more positive the EA of art students will be. Therefore, the following hypothesis is proposed:

**H3:** OB significantly positively influences the EA of art university students.

EM refers to an individual's beliefs and expectations about pursuing entrepreneurship [58]. Previous studies have found that EM significantly influences EI [59–61]. For example, based on social cognitive theory and ecosystem theory, Chahal, Shoukat [60] explored the factors influencing EI among university students, and the results showed that EM significantly positively affects EI. Minh Hue, Thao [61] employed SEM to analyze self-reported data from 341 university students, and their findings indicated that EM directly influences EI. This study hypothesizes that the stronger the EM among art university students, the higher their EI. Based on these findings, the following hypothesis is proposed:

**H4:** EM significantly positively influences the EI of art university students.

Several studies have used the TPB to explore the predictors of EI, and the results consistently show that EA, SN, and PBC are significant predictors of EI [38,62–64]. For example, based on the TPB, Aliedan, Elshaer [63] investigated the impact of university education support on students' EI in Saudi Arabia. They found that EA, SN, and PBC significantly influenced EI. Mensah, Khan [38] used SPSS to analyze self-reported data from 478 university students and tested the significant factors influencing EI. Their findings indicated that EA, SN, and PBC were all significant factors influencing EI. This study hypothesizes that the stronger the EA, SN, and PBC among art university students, the higher their EI. Based on the above research, the following hypotheses are proposed:

**H5:** EA significantly positively influences the EI of art university students.

**H6:** SN significantly positively influences the EI of art university students.

**H7:** PBC significantly positively influences the EI of art university students.

Based on the above hypotheses, the hypothesized model of this study is presented (Fig 3).

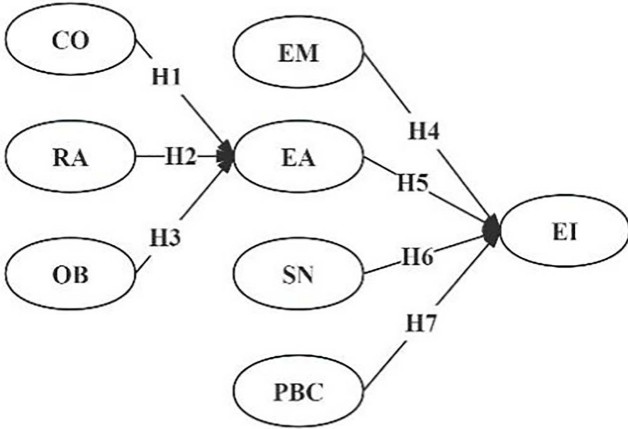

**Fig 3. Hypothesis model.**

## 3. Methodology

### 3.1. Samples

This study was approved by the Ethics Committee of Guangdong University of Petrochemical Technology, with the approval number: GUPT-2024-01-0010. The approval is valid from January 12, 2024, to January 12, 2027. Participants were selected based on the following criteria: (1) they were full-time undergraduate students enrolled in art-related programs (e.g., design, fine arts, media arts) at accredited Chinese universities; (2) they had completed at least one academic year to ensure some exposure to entrepreneurship-related content or activities; and (3) they voluntarily agreed to participate in the study and completed the full questionnaire. Before completing the questionnaire, participants were provided detailed information about the study, including its purpose and privacy protection measures. They were required to give written informed consent before proceeding with the survey, and they had the right to withdraw at any time. The survey was conducted on the Wenjuanxing platform (https://www.wjx.cn/) from April to June 2024. A snowball sampling method was employed during data collection to ensure the quantity and quality of responses. Participants were encouraged to refer other eligible respondents to participate in the survey. Additionally, the time-tracking feature of the Wenjuanxing platform was used to monitor and assess the quality of the data. Respondents who completed the survey and provided valid answers were rewarded with a random cash incentive.

After making the necessary preparations, we collected data from 312 undergraduate students at three universities in Liaoning Province in northeastern China. To ensure the quality of the sample, we followed the method outlined by previous studies and applied three criteria to screen the questionnaires. First, based on a preliminary test, it was determined that under normal circumstances, participants would take 2–5 minutes to complete the questionnaire. Participants who took less than 90 seconds to complete the survey were considered to have rushed through the questionnaire irresponsibly, and their data were deemed invalid. Second, the questionnaire included a reverse-coded item. Participants who provided non-reversed answers to this item were considered to have answered carelessly, and their data were excluded. Lastly, any questionnaires with identical responses were also excluded from the analysis. After rigorous screening, 39 invalid questionnaires were removed, leaving 273 valid responses for further analysis. In the research model of this study, the maximum number of arrows pointing to endogenous latent constructs was four. According to Hair [65], to achieve an $R^2$ of 0.10 in model explanatory power, the minimum sample size required at a 1% significance level is 191. This study used a sample of 273, which exceeds the minimum requirement, ensuring the research findings' robustness and reliability.

Among the 273 valid questionnaires, 86 were completed by male students (31.5%) and 187 by female students (68.5%). In terms of grade, 84 were freshman (30.8%), 62 were sophomore (22.7%), 56 were junior (20.5%), and 71 were senior (26%). 41 students (15%) had entrepreneurial experience, while 232 students (85%) had no prior experience. Regarding entrepreneurship education, 118 students (43.2%) reported having participated in at least one form of entrepreneurship education, including formal university courses, workshops, lectures, or extracurricular activities related to entrepreneurship. In comparison, 155 students (56.8%) indicated that they had not participated in such activities (Table 1). This representative sample provides a solid foundation for analyzing the relationships between EM, RA, CO, OB, EA, SN, PBC, and EI of art university students. The diversity and size of the sample help ensure that the research findings are widely applicable to the broader student population in this context.

### 3.2. Measurements

The measurements consist of two main sections: the first section collects participants' demographic information, while the second section gathers self-reported data on various constructs. Validated scales were used to assess these constructs, which were adjusted to align with the research context and objectives. This approach ensures the tools accurately reflect participants' conditions in the research setting.

In addition to basic demographic information, the measurements include eight key constructs: RA, CO, OB, EM, EA, SN, PBC, and EI. Compared to a 7-point Likert scale, a 5-point Likert scale has significant advantages in improving scale reliability and validity [66], reducing bias, and enhancing the sensitivity of statistical analysis [67]. Therefore, each construct was measured using a 5-point Likert scale, with responses ranging from (1) "Completely disagree" to (5) "Completely agree." (Table 2)

The RA scale measures how art university students perceive that new entrepreneurial ideas, products, or services have advantages over traditional options. The scale consists of four items and demonstrates excellent reliability with a Cronbach's alpha 0.940.

The CO scale measures the degree to which entrepreneurial ideas, business models, products, or services align with existing values, experiences, and needs. The scale consists of three items and demonstrates good reliability with a Cronbach's alpha 0.897.

The OB scale measures how others perceive, observe, or evaluate innovative outcomes of entrepreneurial activities, products, or services. The scale consists of four items and demonstrates good reliability with a Cronbach's alpha 0.819.

The EM scale measures individuals' beliefs and expectations regarding the personal outcomes of pursuing entrepreneurship. The scale consists of 10 items and demonstrates good reliability with a Cronbach's alpha 0.820.

**Table 1. Demographic information of the sample.**

| Demographic Information | Category | Frequency | Percentage |
|---|---|---|---|
| Gender | Male | 86 | 31.5% |
| | Female | 187 | 68.5% |
| Grade | Freshman | 84 | 30.8% |
| | Sophomore | 62 | 22.7% |
| | Junior | 56 | 20.5% |
| | Senior | 71 | 26.0% |
| Entrepreneurial Experience | Yes | 41 | 15.0% |
| | No | 232 | 85.0% |
| Entrepreneurship Education | Yes | 118 | 43.2% |
| | No | 155 | 56.8% |

**Table 2. Source and example items of scales.**

| Scale | Source | Example item |
|---|---|---|
| RA | Al-Rahmi, Yahaya [68] | I believe entrepreneurship positively impacts our school. |
| CO | | Entrepreneurial practice activities are compatible with my entrepreneurial ideas. |
| OB | | When students see entrepreneurial projects being promoted, they show great interest. |
| EM | Yi and Duval-Couetil [69] | Entrepreneurship allows me to focus on the technology I am most interested in. |
| EA | Liu, Gorgievski [70] | I would like to start a company if I had the opportunity and resources. |
| SN | | My closest family members think I should start a business. |
| PBC | | Starting and maintaining a company would be easy for me. |
| EI | Dwi Lestari, Rizkalla [71] | I am willing to do anything related to becoming an entrepreneur. |

The EA scale measures art students' positive or negative evaluations of entrepreneurship. The scale comprises five items and demonstrates excellent reliability with a Cronbach's alpha 0.940.

The SN scale measures how much others influence art students' EI. The scale consists of three items and demonstrates excellent reliability with a Cronbach's alpha 0.910.

The PBC scale is designed to measure art students' evaluations of the ease or difficulty of entrepreneurship. The scale comprises six items and demonstrates excellent reliability with a Cronbach's alpha 0.950.

The EI scale measures individuals' tendencies and intentions toward entrepreneurial behavior based on intrinsic motivation and external environmental factors. The scale comprises six items and demonstrates excellent reliability with a Cronbach's alpha 0.928.

### 3.3. Data analysis

This study employs a two-stage approach to test the hypotheses and build the predictive model. First, PLS-SEM is used to identify the linear relationships between exogenous and endogenous variables. This study employed PLS-SEM rather than Covariance-Based SEM. First, PLS-SEM is more appropriate for exploratory research models emphasizing prediction and theory development, rather than theory testing and model fit assessment [65]. Given that our study integrates constructs from two theoretical frameworks (DOI and TPB), PLS-SEM offers greater flexibility in handling such complex models. Second, PLS-SEM is well-suited for small to medium sample sizes and does not require the assumption of multivariate normality, making it appropriate for our dataset (n = 273). Lastly, PLS-SEM is particularly effective in prioritizing predictor variables based on their contribution to explained variance ($R^2$), which aligns closely with our goal of identifying the most significant predictors of EI. However, PLS-SEM has limitations in capturing non-linear and non-compensatory relationships, which are crucial for understanding the factors influencing EI among art students.

To address these limitations, the second stage of this study integrates ANN. ANN can capture linear and non-linear relationships using non-compensatory models, enhancing predictive accuracy [72,73]. Moreover, previous research has highlighted the robustness of ANN in handling complex data patterns and predictive tasks [74]. This study deepens the analysis by combining SEM and ANN. It improves accuracy, providing a comprehensive understanding of the factors influencing EI among art students and their relative importance.

### 3.4. Rationale for selecting theoretical variables

In this study, three core variables—RA, CO, and OB—were selected from the five original dimensions of the DOI theory. The main reasons are as follows: First, for the sake of maintaining a parsimonious model structure, not all DOI variables were included. Incorporating Complexity and Trialability would significantly increase the number of model paths and computational burden, potentially reducing model robustness, especially given the limited sample size in this study [75].

Second, Complexity and Trialability often pose measurement challenges in entrepreneurship research within university settings. The interpretation of Complexity can vary greatly across individuals, leaving room for ambiguity; Trialability, on the other hand, is typically difficult to realize in practice at the student level, making it less actionable [75]. Finally, prior studies have shown that RA, CO, and OB are the most representative and predictive constructs within DOI theory, and they have been widely applied in studies related to EA [76]. Therefore, this study prioritizes these three variables to balance theoretical representativeness and model feasibility.

## 4. Results

Researchers employed various statistical methods to develop and validate the research findings. Hair [77] differentiated between the applications of first-generation and second-generation statistical methods. Factor analysis and regression analysis dominated first-generation statistical methods and were widely used. Since the 1990s, more sophisticated multivariate statistical methods, such as SEM, have become second-generation [78]. There are two types of SEM: covariance-based SEM and variance-based SEM. In this study, due to the complexity of the model—including eight constructs, 48 items, and seven relationships—PLS-SEM was deemed appropriate for analyzing such a complex model [77]. SmartPLS 4.0 was used to test both the measurement and structural models.

### 4.1. Measurement model

To evaluate the measurement model, this study followed the standard procedures recommended by Hair [65]. Indicator reliability was first examined through the outer loadings of each item. Loadings above 0.70 were considered acceptable, while items with loadings between 0.40 and 0.70 were retained only if their removal would not improve the composite reliability (CR) and average variance extracted (AVE). Internal consistency reliability was assessed using both Cronbach's alpha and CR values, all exceeding the recommended threshold of 0.70, indicating satisfactory reliability (Table 3).

Third, the correlation matrix for the Fornell-Larcker discriminant validity test is presented in Table 4. According to Hair [65], the square root of the AVE for each construct should be higher than the highest correlation between that construct and any other construct in the model. The results meet this criterion.

HTMT, proposed by Henseler, Ringle [79], is a criterion for assessing discriminant validity. HTMT is calculated as the ratio of the average heterotrait-heteromethod correlations to the average monotrait-heteromethod correlations. Heterotrait-heteromethod correlations measure the relationships between indicators across different constructs, while monotrait-heteromethod correlations measure the relationships between indicators within the same construct. The HTMT values were calculated using SmartPLS software (Table 5). All HTMT values fall within the acceptable threshold of ≤0.90 [79].

Harman's single-factor test was conducted using SPSS to statistically assess common method bias. All items were entered into an exploratory factor analysis with unrotated principal component extraction. The results indicated that the first factor accounted for 32.4% of the total variance, below the 50% threshold, suggesting that common method bias is not a serious concern in this study [80].

### 4.2. Structural model

The structural model was evaluated following the steps recommended by Hair [65]:

(1) Assess collinearity issues in the structural model (VIF < 5);

(2) Evaluate the significance and relevance of structural model relationships (p < 0.05);

(3) Assess the coefficient of determination ($R^2$) level:Thresholds: 0.190 indicates a weak level, 0.333 indicates a moderate level, and 0.670 indicates a strong level.

**Table 3. Reliability and AVE.**

| Constructs | Items | Outer loadings | Cronbach's α | CR | AVE |
|---|---|---|---|---|---|
| EA | EA1 | 0.517 | 0.807 | 0.845 | 0.570 |
| | EA2 | 0.833 | | | |
| | EA3 | 0.728 | | | |
| | EA4 | 0.826 | | | |
| | EA5 | 0.824 | | | |
| CO | CO1 | 0.862 | 0.863 | 0.864 | 0.786 |
| | CO2 | 0.923 | | | |
| | CO3 | 0.873 | | | |
| EI | EI1 | 0.618 | 0.905 | 0.919 | 0.685 |
| | EI2 | 0.863 | | | |
| | EI3 | 0.765 | | | |
| | EI4 | 0.904 | | | |
| | EI5 | 0.873 | | | |
| | EI6 | 0.906 | | | |
| EM | EM1 | 0.831 | 0.807 | 0.809 | 0.722 |
| | EM2 | 0.871 | | | |
| | EM3 | 0.846 | | | |
| OB | OB1 | 0.465 | 0.730 | 0.763 | 0.517 |
| | OB2 | 0.845 | | | |
| | OB3 | 0.683 | | | |
| | OB4 | 0.821 | | | |
| PBC | PBC1 | 0.718 | 0.845 | 0.849 | 0.564 |
| | PBC2 | 0.718 | | | |
| | PBC3 | 0.794 | | | |
| | PBC4 | 0.730 | | | |
| | PBC5 | 0.771 | | | |
| | PBC6 | 0.771 | | | |
| RA | RA1 | 0.764 | 0.874 | 0.890 | 0.726 |
| | RA2 | 0.859 | | | |
| | RA3 | 0.883 | | | |
| | RA4 | 0.896 | | | |
| SN | SN1 | 0.813 | 0.851 | 0.860 | 0.771 |
| | SN2 | 0.907 | | | |
| | SN3 | 0.911 | | | |

First, collinearity issues were assessed using the Variance Inflation Factor (VIF). A VIF value ≥ 5 indicates potential collinearity problems. All VIF values in this study were within the acceptable threshold (VIF < 3) (Table 6). Therefore, there were no collinearity issues in this study.

Second, the path coefficients (β) for the relationships between constructs in the model are shown in Table 7. The significance of the path coefficients was evaluated using the bootstrapping algorithm in PLS. The t-values and p-values were used to determine whether the β is statistically significant at the 5% significance level. A 5% significance level indicates that p-values must be less than 0.05 and t-values must exceed 1.96. The results of the bootstrapping algorithm are presented in Table 7. Among the direct significant predictors of EI, PBC had the strongest effect (β = 0.343, t = 7.234, p = 0.000), followed by EA (β = 0.323, t = 6.566, p = 0.000), SN (β = 0.222, t = 4.028, p = 0.000), and EM (β = 0.102, t = 2.183,

**Table 4. Discriminant validity (Fornell-Larcker Criteria).**

| Constructs | EA | CO | EI | EM | OB | PBC | RA | SN |
|---|---|---|---|---|---|---|---|---|
| EA | **0.755** | | | | | | | |
| CO | 0.359 | **0.887** | | | | | | |
| EI | 0.628 | 0.557 | **0.828** | | | | | |
| EM | 0.414 | 0.343 | 0.412 | **0.850** | | | | |
| OB | 0.375 | 0.660 | 0.579 | 0.301 | **0.719** | | | |
| PBC | 0.478 | 0.400 | 0.653 | 0.303 | 0.421 | **0.751** | | |
| RA | 0.467 | 0.672 | 0.483 | 0.479 | 0.574 | 0.351 | **0.852** | |
| SN | 0.444 | 0.280 | 0.591 | 0.326 | 0.304 | 0.560 | 0.229 | **0.878** |

Note: The bolded values on the diagonal represent each construct's square root of the AVE.

**Table 5. Discriminant validity (HTMT Criteria).**

| | EA | CO | EI | EM | OB | PBC | RA | SN |
|---|---|---|---|---|---|---|---|---|
| EA | | | | | | | | |
| CO | 0.422 | | | | | | | |
| EI | 0.705 | 0.637 | | | | | | |
| EM | 0.518 | 0.410 | 0.489 | | | | | |
| OB | 0.372 | 0.823 | 0.669 | 0.332 | | | | |
| PBC | 0.553 | 0.463 | 0.732 | 0.365 | 0.513 | | | |
| RA | 0.545 | 0.772 | 0.541 | 0.567 | 0.676 | 0.394 | | |
| SN | 0.514 | 0.319 | 0.666 | 0.396 | 0.338 | 0.661 | 0.253 | |

**Table 6. VIF.**

| Construct | VIF |
|---|---|
| CO | 2.292 |
| RA | 1.929 |
| OB | 1.874 |
| EM | 1.250 |
| EA | 1.502 |
| SN | 1.575 |
| PBC | 1.617 |

**Table 7. Results of hypothesis testing.**

| Hypothesis | β | Standard deviation | T Statistics | P Values | Results |
|---|---|---|---|---|---|
| EA→EI | 0.323 | 0.049 | 6.566 | 0.000 | Supported |
| PBC→EI | 0.343 | 0.047 | 7.234 | 0.000 | Supported |
| EM→EI | 0.102 | 0.047 | 2.183 | 0.029 | Supported |
| SN→EI | 0.222 | 0.055 | 4.028 | 0.000 | Supported |
| OB→EA | 0.158 | 0.089 | 1.781 | 0.075 | Not Supported |
| RA→EA | 0.373 | 0.073 | 5.131 | 0.000 | Supported |
| CO→EA | 0.004 | 0.101 | 0.044 | 0.965 | Not Supported |

p = 0.029). Meanwhile, only RA positively influenced EA (β = 0.373, t = 5.131, p = 0.000). However, OB (β = 0.158, t = 1.781, p = 0.075) and CO (β = 0.004, t = 0.044, p = 0.965) did not significantly affect EA.

Third, the coefficient of determination ($R^2$) represents the proportion of variance in an endogenous construct explained by all related exogenous constructs [65]. Values around 0.67 are considered substantial, around 0.33 are moderate, and around 0.19 are weak. As shown in Table 8, EM, EA, SN, and PBC collectively explained 60% of the variance in EI.

### 4.3. Artificial neural network analysis

Given the potential non-linear relationships between exogenous and endogenous variables, this study uses the significant factors from the SEM-PLS path analysis as input neurons for the ANN model. The rationale for applying ANN includes the non-normal distribution of data and ANN's robustness to noise, outliers, and small sample sizes. Additionally, ANN is suitable for non-compensatory models, where a decrease in one factor does not need to be compensated by an increase in another. The ANN analysis was implemented using IBM's SPSS neural network module. The ANN algorithm can capture linear and non-linear relationships and does not require the data to follow a normal distribution [81]. The algorithm learns through training and uses the feedforward-backpropagation (FFBP) algorithm to predict outcomes [82]. Multilayer perceptrons and sigmoid activation functions were used for the input and hidden layers [83]. Through multiple iterations of the learning process, errors can be minimized, further improving prediction accuracy [84].In this study, 70% of the sample was used for training, while the remaining 30% was used for testing. To avoid the possibility of overfitting, a ten-fold cross-validation procedure was performed, and the root mean square error (RMSE) was calculated [85]. As shown in Table 9, the average RMSE values for the training and testing processes were 0.1910 and 0.1860, respectively, confirming that the model achieved an excellent fit.

To evaluate the predictive capability of each input neuron, this study conducted a sensitivity analysis (Table 10). The normalized importance of each neuron was calculated by dividing its relative importance by the highest importance value and expressing the result as a percentage [86]. The results of the sensitivity analysis revealed that PBC was the most influential predictor (100%), followed by EA (70.8%), SN (57.6%), RA (43.1%), and EM (31.2%). These normalized importance values indicate the relative weight of each predictor in determining EI in the ANN model. The ranking suggests that PBC and EA play dominant roles under linear and non-linear assumptions. The implications of these results are further discussed in Section 5.

## 5. Discussion

This study explored the factors influencing EI among art university students. Based on a comprehensive literature review, this study hypothesized that EM, EA, SN, and PBC would significantly predict EI, and that RA, CO, and OB would significantly influence EA. First, the validity of the proposed research model was evaluated using Smart PLS software. The results supported all hypotheses except for the relationships between CO and EA, as well as OB and EA, which were found to be insignificant. Collectively, all predictor variables explained 60% of the total variance in EI. Second, a ten-fold cross-validation and sensitivity analysis were conducted using ANN. The results indicated that PBC was the most important predictor, followed by EA, SN, RA, and EM. The detailed discussion of these findings concerning the initially proposed research questions and hypotheses is presented below.

PBC significantly and positively influences the EI of art university students. This finding suggests that the stronger their confidence and perceived ability to engage in entrepreneurial activities, the higher their EI. In the Chinese context, many

**Table 8. $R^2$.**

| Constructs | $R^2$ |
|---|---|
| EA | 0.235 |
| EI | 0.600 |

**Table 9. Root mean square of error values.**

| Training | | | Testing | | | Total samples |
|---|---|---|---|---|---|---|
| N | SSE | RMSE | N | SSE | RMSE | |
| 187 | 7.0958 | 0.1948 | 86 | 3.4506 | 0.2003 | 273 |
| 196 | 7.6752 | 0.1979 | 77 | 1.6974 | 0.1485 | 273 |
| 181 | 6.3726 | 0.1876 | 92 | 3.7614 | 0.2022 | 273 |
| 190 | 6.6078 | 0.1865 | 83 | 2.5904 | 0.1767 | 273 |
| 199 | 6.8672 | 0.1858 | 74 | 2.6636 | 0.1897 | 273 |
| 198 | 6.7806 | 0.1851 | 75 | 3.1228 | 0.2041 | 273 |
| 194 | 6.7446 | 0.1865 | 79 | 2.9978 | 0.1948 | 273 |
| 191 | 7.0554 | 0.1922 | 82 | 2.5484 | 0.1763 | 273 |
| 193 | 6.216 | 0.1795 | 80 | 2.7146 | 0.1842 | 273 |
| 181 | 8.3378 | 0.2146 | 92 | 3.0952 | 0.1834 | 273 |
| Mean | 6.9753 | 0.1910 | Mean | 2.8642 | 0.1860 | |
| Sd | | 0.0098 | Sd | | 0.0166 | |

Note: N: number of samples; SSE: sum of squares of error; RMSE: root mean square of error

**Table 10. Sensitivity analysis.**

| Artificial neural network (ANN) | EA | SN | EM | PBC | RA |
|---|---|---|---|---|---|
| ANN1 | 0.669 | 0.349 | 0.331 | 1.000 | 0.377 |
| ANN2 | 0.570 | 0.567 | 0.216 | 1.000 | 0.366 |
| ANN3 | 0.854 | 0.795 | 0.218 | 1.000 | 0.480 |
| ANN4 | 0.705 | 0.501 | 0.279 | 1.000 | 0.390 |
| ANN5 | 0.688 | 0.481 | 0.154 | 1.000 | 0.389 |
| ANN6 | 0.756 | 0.560 | 0.288 | 1.000 | 0.431 |
| ANN7 | 0.824 | 0.582 | 0.214 | 1.000 | 0.413 |
| ANN8 | 0.915 | 0.758 | 0.346 | 1.000 | 0.449 |
| ANN9 | 0.724 | 0.558 | 0.318 | 1.000 | 0.483 |
| ANN10 | 0.373 | 0.605 | 0.759 | 1.000 | 0.532 |
| Mean importance | 0.708 | 0.576 | 0.312 | 1.000 | 0.431 |
| Normalized importance (%) | 70.8% | 57.6% | 31.2% | 100.0% | 43.1% |

art universities emphasize practical, project-based learning, participation in national design competitions (e.g., "Internet+ Innovation and Entrepreneurship Competition"), and portfolio development. These experiences enhance students' entrepreneurial self-efficacy by exposing them to real-world challenges and enabling them to showcase their talents [87]. Furthermore, the growing number of school-enterprise collaborations and creative industry incubators in Chinese institutions provides fertile ground for students to gain practical experience and build confidence in their entrepreneurial competencies. These institutional characteristics contribute to stronger PBC among art students, enhancing their EI.

EA significantly and positively influences the EI of art university students. A more favorable attitude toward entrepreneurship leads to a higher EI. In recent years, Chinese art universities have actively promoted entrepreneurship education by incorporating interdisciplinary curricula, offering specialized courses in art and design entrepreneurship, and establishing creative incubation centers. Government-supported initiatives such as the "Double Innovation" policy (mass entrepreneurship and innovation) have further encouraged universities to build ecosystems that stimulate creative self-employment [88]. For example, some universities integrate studio practice with market-oriented projects, allowing students to develop

creative and entrepreneurial thinking. These localized educational strategies shape students' positive attitudes toward entrepreneurship by demonstrating its feasibility and relevance to their artistic aspirations.

SN significantly and positively influences the EI of art university students. This result suggests that social support and normative expectations are crucial in shaping EI. In collectivist societies like China, family, mentors, and peer networks influence students' career decisions. Many Chinese families view entrepreneurship positively—especially in creative fields—because it reflects initiative, independence, and potential economic success [89]. Moreover, Chinese art students often engage in creative communities, participate in exhibitions, and connect with alumni entrepreneurs, enhancing their exposure to entrepreneurial role models. These supportive social environments help students internalize entrepreneurship as a desirable and achievable career path, boosting their intention to pursue it.

EM significantly and positively influences the EI of art university students. This indicates that the stronger their intrinsic drive to become entrepreneurs, the higher their EI. In China, many art students are motivated by the desire for creative autonomy, cultural expression, and the opportunity to develop a personal brand. The expansion of the cultural and creative industries—such as independent design, animation, and digital content—has created more entrepreneurial pathways that align with students' passions and identities. These motivations are economic and deeply rooted in personal fulfillment and aesthetic values, distinguishing art students from their peers in more traditional disciplines. Therefore, their EM stems from self-expression and professional ambition [87], reinforcing their EI.

RA significantly and positively influences the EA of art university students. When students perceive entrepreneurship as offering greater autonomy, creative freedom, and alignment with their personal goals, they are more likely to develop a positive attitude toward it. In the Chinese art education system, entrepreneurship is often framed not just as a financial activity but as a form of artistic independence and self-realization [89]. Unlike conventional employment, starting a business allows students to retain ownership of their intellectual and creative output. This appeal is particularly strong among students pursuing visual design, animation, or digital media, where personal style and originality are marketable assets. The perceived RA of entrepreneurship in these fields thus contributes to a more favorable EA.

CO and OB were not found to significantly influence the EA of art university students. This contradicts earlier findings [53,57] and may be due to the unique mindset of students in creative disciplines. For many Chinese art students, pursuing entrepreneurship is not solely based on practical compatibility or observable success. However, it is shaped by deeper values such as personal meaning, identity expression, and long-term artistic fulfillment. These factors may outweigh the perceived congruence of entrepreneurial activities with their current experiences or the visibility of others' entrepreneurial success. Additionally, some students may perceive showcased entrepreneurial cases as overly commercialized or not representative of authentic creative work, reducing the impact of OB on their attitude formation.

The results of the ANN analysis provided richer insights compared to the linear SEM approach. Although both methods identified PBC and EA as key predictors, the ANN emphasized that the importance of RA and EM was relatively lower when nonlinear effects were considered. This suggests that although these variables were statistically significant in the SEM analysis, their actual impact may be more context-dependent or mediated by non-compensatory mechanisms. Moreover, by incorporating the ANN approach, this study addresses the potential limitations of SEM in capturing complex, nonlinear relationships. ANN can detect hidden patterns and prioritize predictors, offering a more nuanced understanding of how EI is formed. This hybrid approach enhances the robustness of the findings and provides practical guidance for prioritizing intervention strategies.

### 5.1. Theoretical implications

First, this study focuses specifically on the entrepreneurial behavior of art university students, marking its first theoretical significance. Existing entrepreneurship research predominantly centers on general university students or those in specific disciplines, such as business administration, with a limited in-depth exploration of art university students. Art students possess unique creative thinking and artistic sensitivity, and various factors, including personal artistic interests, creativity,

sociocultural contexts, and educational resources, influence their EI. Thus, this study addresses a theoretical gap in the research on the EI of art university students and provides theoretical guidance for entrepreneurship education in art institutions.

Second, the integration of the DOI and the TPB constitutes the second theoretical significance of this study. By combining these two frameworks, the study provides a novel theoretical model for understanding the EI of art university students. The DOI focuses on disseminating technologies, innovations, or new ideas, while the TPB emphasizes the psychological motivations and intentions behind individual behaviors. This integration enables a more comprehensive analysis of the interaction between the diffusion of innovative information and individual behavioral intentions when art university students encounter entrepreneurial opportunities, thereby addressing a theoretical gap in the existing literature on the entrepreneurial behavior of this unique group.

Third, the integration of SEM and ANN techniques represents the third theoretical significance of this study. The study introduces a hybrid methodological approach combining these two advanced data analysis methods. SEM facilitates the verification of causal relationships among variables and the evaluation of model fit, while ANN effectively handles non-linear relationships and complex data, enhancing predictive accuracy. This methodological integration allows the study to more precisely identify the key factors influencing the EI of art university students and provides robust theoretical evidence, offering valuable data support for future research and practical applications.

Fourth, the study's findings reveal that CO and OB have no significant relationship with EA, challenging the traditional DOI's universal applicability in certain contexts. This result prompts a reevaluation of the theory's application to the entrepreneurial domain of art university students. It suggests a more nuanced examination of how different variables perform across various groups and the underlying reasons for these differences. This discovery provides new insights for future research, contributing to the enrichment and refinement of the theoretical framework on factors influencing EI. Furthermore, it offers more targeted policy recommendations and practical guidance to support entrepreneurship among art university students.

Finally, this study extends the theoretical understanding of EI by highlighting its unique characteristics in creative fields compared to technical or business contexts. In business and technical domains, EI is often driven by rational opportunity evaluation, market needs, and profit-maximizing behavior [90]. By contrast, EI in creative domains such as art and design is more deeply influenced by intrinsic motivations, self-expression, aesthetic values, and the pursuit of cultural or social impact [91]. Creative entrepreneurship is often identity-driven and may be less responsive to traditional predictors like financial incentives or market competition. This divergence suggests that existing entrepreneurship models should be adapted to account for value-driven, emotion-laden, and identity-related factors more common among creative individuals. Our findings support this by showing that constructs such as EA and EM—rather than purely rational evaluations—play stronger roles in shaping EI in the art student population.

## 5.2. Practical implications

This study provides an in-depth exploration of the factors influencing EI among art university students. The findings hold significant practical value for understanding and promoting EI among university students.

Universities: First, given that PBC is the strongest predictor of EI, universities should prioritize establishing a comprehensive entrepreneurship education system. This system should ensure that students gain extensive entrepreneurial knowledge, skills, and practical experience. Measures include offering diverse entrepreneurship courses, funding support, and creating business incubators to enhance students' practical capabilities and confidence. Second, recognizing that EA is a key driver of EI, universities should foster a strong entrepreneurial culture through various initiatives. Hosting entrepreneurship competitions, lectures, and networking events like entrepreneurship salons can ignite students' entrepreneurial passion. Sharing the success stories of entrepreneurs can also help students cultivate a positive attitude toward entrepreneurship. Finally, considering the critical role of SN in EI, universities should implement a mentorship program.

                                                         

This program could invite experienced entrepreneurs, investors, and other professionals as mentors to provide personalized guidance to students. Such mentorship can strengthen students' sense of SN by fostering the perception of expectations and support from significant others, thereby enhancing their EI.

Teachers: First, since PBC is the strongest predictor of EI, teachers should focus on enhancing students' entrepreneurial skills and capabilities. Through case studies, simulated entrepreneurship exercises, and workshops, teachers can help students develop critical skills, including market analysis, financial management, and project management. Strengthening these practical abilities and confidence will enhance students' PBC. Second, recognizing that EA is a key driver of EI, teachers should actively share stories of successful entrepreneurs and provide updates on industry trends to inspire students' positive perceptions of entrepreneurship. By fostering critical thinking skills, teachers can help students identify and evaluate the RA of entrepreneurial opportunities—key antecedents of attitude formation—further reinforcing their positive EA. Finally, considering the importance of SN in EI, teachers can organize interdisciplinary team projects, encouraging collaboration among students from diverse backgrounds to broaden their perspectives and enhance teamwork skills. Moreover, they should guide students in building extensive social networks that include alumni, industry experts, and other key stakeholders, enabling them to access greater support and resources during their entrepreneurial journey.

Students: First, given that PBC is the strongest predictor of EI, students should deeply understand their interests, strengths, and career aspirations to clarify their entrepreneurial direction and goals. This self-awareness enables students to better assess their entrepreneurial capabilities and resources, enhancing their confidence in overcoming challenges and obstacles in the entrepreneurial process. Second, recognizing that EA is a key driver of EI, students should stay informed about industry trends and identify the RA of entrepreneurial opportunities, such as market demand and technological innovation. Engaging in entrepreneurship training, reading relevant books, and exploring success stories can help students cultivate a positive perception and attitude toward entrepreneurial activities. Finally, as motivation significantly influences behavioral intentions, students should be willing to experiment with new ideas and face the failures and setbacks inherent in entrepreneurship. They should also remain flexible and adaptive, adjusting their strategies based on market feedback and personal circumstances. This proactive and adaptive approach, driven by intrinsic motivation, fosters resilience and continuous progress along the entrepreneurial journey.

## 5.3. Limitations and future research

Despite its theoretical and practical significance, this study has certain limitations. First, the cultural context of China may influence how art university students form EI. In collectivist societies like China, family expectations, social harmony, and Confucian values such as responsibility and perseverance can shape students' motivations differently from those in more individualistic cultures [92]. As a result, the findings of this study may not be fully generalized to other cultural settings. Future studies are encouraged to conduct cross-cultural comparisons to better understand how cultural values affect EI in creative fields. Second, this study adopts a cross-sectional design, collecting data at a single point in time. As a result, it cannot capture the dynamic changes in EI and their influencing factors. The formation and evolution of EI may change over time as individuals gain experience. Future studies could employ longitudinal designs to track the developmental trajectory of EI among art university students, providing deeper insights into their influencing factors and mechanisms. Third, the sample composition in this study consists of 68.5% male students and 31.5% female students, leading to a gender imbalance. This imbalance may introduce potential biases in the analysis of gender differences, affecting the generalizability and representativeness of the findings. Future research should strive to achieve a more balanced gender ratio or explicitly account for gender as a variable in the analysis to ensure more comprehensive and accurate conclusions. Finally, one notable limitation of this study is convenience sampling from only three art universities in China. While the sample offers valuable insights into the EI of art students, the findings may not be fully generalizable to other disciplines, regions, or educational systems. Future research should aim to include more diverse and representative samples, potentially using stratified or random sampling techniques across a broader range of institutions.

## 6. Conclusion

This study, grounded in the DOI and the TPB, investigated the factors influencing the EI of art university students by combining SEM and ANN. Data were collected from 273 art university students, and seven hypotheses were empirically tested. The findings revealed that, among the predictors of EI, PBC emerged as the strongest predictor, followed by EA, SN, and EM. Additionally, RA was identified as an antecedent of EA, whereas CO and OB showed no significant effect on EA. Moreover, during the ten-fold cross-validation of the ANN, the model demonstrated a good fit, and the predictive strength of individual constructs was re-examined through sensitivity analysis. This study provides a new perspective on cultivating EI among art university students through education and teaching practices while contributing a new dimension to the theoretical framework of EI. Furthermore, it offers targeted recommendations for enhancing EI from three levels: universities, teachers, and students.

## Supporting information

**S1 Data. Raw Data.**
(XLSX)

## Author contributions

**Conceptualization:** Yiliang Cao.

**Formal analysis:** Yiliang Cao.

**Methodology:** Jie Zhang.

**Supervision:** Jie Zhang.

**Writing – original draft:** Jie Zhang.

**Writing – review & editing:** Yiliang Cao.

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
