## [Decision Letter · Decision Letter 0]

28 Apr 2025

PONE-D-24-60132Factors Influencing Entrepreneurial Intention among Art University Students: Based on Innovation Diffusion Theory and the Theory of Planned BehaviorPLOS ONE

Dear Dr. Zhang,

Thank you for submitting your manuscript to PLOS ONE. After careful consideration, we feel that it has merit but does not fully meet PLOS ONE’s publication criteria as it currently stands. Therefore, we invite you to submit a revised version of the manuscript that addresses the points raised during the review process.

The paper represents a valuable contribution to the study of entrepreneurial intentions in the arts. It is based on appropriate theory and adequately utilizes appropriate research methods. However, summarizing the comments of the three reviewers and my observations, the manuscript needs some more work to reach the level for publication in PLOS ONE. Primarily, this refers to the improvement of some elements of methodology and discussion, as well as certain technical aspects. Please take into account the comments of the reviewers and editor and submit a new version of the manuscript.Editor's comments:

The title should be shortened - I suggest omitting the second part of the title, since combining TPB and DOI in research of this type is not newI suggest proofreading the paper to remove spelling and stylistic language errorsAMOS is the name of the software, not the data analysis method so that I would exclude it from the list of methods in the introductionFigure 1 and Figure 2 should come with the source/reference since the models are not originally yoursIt is necessary to provide insight into the items by which the variables were measured, along with referencesWhat is shown in Table 5? It is not mentioned in the text at allThe discussion should be more refined, enriched by a better elaboration of the results' similarities and differences with other comparable studies, and with more theoretical implications

We look forward to receiving your revised manuscript.

Kind regards,

Tamara Šmaguc, Ph.D

Academic Editor

PLOS ONE

3. In the online submission form, you indicated that [The data that support the findings of this study are available on request from the corresponding author.].

Reviewers' comments:

Reviewer's Responses to Questions

**Comments to the Author**

1. Is the manuscript technically sound, and do the data support the conclusions?

Reviewer #1: Yes

Reviewer #2: Yes

Reviewer #3: Yes

2. Has the statistical analysis been performed appropriately and rigorously? 

Reviewer #1: Yes

Reviewer #2: Yes

Reviewer #3: Yes

3. Have the authors made all data underlying the findings in their manuscript fully available?

Reviewer #1: Yes

Reviewer #2: Yes

Reviewer #3: Yes

4. Is the manuscript presented in an intelligible fashion and written in standard English?

Reviewer #1: Yes

Reviewer #2: Yes

Reviewer #3: Yes

5. Review Comments to the Author

Reviewer #1: Thank you for the opportunity to read your paper. The paper presents valuable insights and makes a meaningful contribution to entrepreneurship research in creative disciplines. The focus on art students adds novelty as most EI research targets business or engineering students. The usage of Theory of planned behavior as well as Innovation Diffusion Theory is appropriate and well-grounded in entrepreneurship literature. Your findings have practical implications and can inform curriculum design and entrepreneurial policy, particularly in creative disciplines. Authors nicely present the problem and research gap, and their paper contribution and objectives. Hypotheses are based on the existing research. Methodology used is appropriate and research results are presented in an understandable manner. Several minor issues are advised:

• The literature review is thorough but could benefit from more international comparative studies (For instance Pejic Bach, M. et al. (2018). Examining determinants of entrepreneurial intentions in Slovenia: applying the theory of planned behaviour and an innovative cognitive style. Economic research-Ekonomska istraživanja, 31(1), 1453-1471.; Hossain, M. I., Tabash, M. I., Siow, M. L., Ong, T. S., & Anagreh, S. (2023). Entrepreneurial intentions of Gen Z university students and entrepreneurial constraints in Bangladesh. Journal of innovation and entrepreneurship, 12(1), 12.

• explain sample selection criteria

• The use of convenience sampling from only three universities may limit generalizability. A discussion of sampling limitations should be more prominent in the Discussion section.

• Common method bias is not address. Adding a procedural or statistical would strengthen the validity.;

• While SEM-PLS is valid, the rationale for using this method over CB-SEM could be briefly elaborated.

• For those of us not familiar with provinces in China, please state in your paper that Liaoning Province is in China

• The discussion restates findings well but could be enriched by more theoretical implications, particularly how EI in creative fields differs from technical/business contexts.

• Consider reflecting on how cultural factors in China may shape EI in ways that might not generalize elsewhere.

• The language is generally clear, though some minor grammatical issues and awkward phrasing appear.

Reviewer #2: Comment 1

In sections 2.1 and 2.2, use full names ("Diffusion of Innovation Theory" and "Theory of Planned Behavior") in titles instead of acronyms.

Comment 2

The explanation of selected variables in 2.1 should be moved to the methods section. Also, please clarify why certain DOI variables were excluded. If complexity is the issue, specify what kind—statistical, theoretical, or interpretative?

Comment 3

In the TPB section of literature review, specify which universities or types of art schools the students come from. Perhaps discussion of the results from the literature regarding differences in TPB when different students are considered (business school, medicine, law...) Also, consider expanding the TPB discussion from the entrepreneurial intention perspective, as there is rich literature in this domain.

Comment 4

The integration of DOI and TPB is conceptually promising. However, before 2.3, clarify how the two theories are merged given their different dependent variables (intention vs. actual use/adoption). Are these considered synonymous? You decided to make EA (attitudes) depended variable for CO, RA and OB, so this will require some elaboration, as DOI suggests that adoption is depended variable. Also, discuss potential multicollinearity between constructs.

Comment 5

The sentence "Regarding entrepreneurial education, 118 students (43.2%) participated, while 155 students (56.8%) did not" is unclear. Participated in what exactly—formal courses, workshops, or something else?

Comment 6

The list of measurement scales before section 3.3 would be clearer in tabular form: Scale | Source | Example item.

Comment 7

Section 4.1 is overly segmented. Instead of listing validation steps in bullet-point style, rewrite the text as a flowing paragraph with light narrative explanation.

Comment 8

Table 5 is mislabeled—VIF is not a correlation matrix. Please present VIF in standard form: one value per predictor.

Comment 9

The ANN results in section 4.3 are not clearly interpreted. The tables are present, but their meaning and implications remain unclear.

Comment 10

In the discussion, you state "At the same time, RA, CO, and OB are important predictors of attitude.", but your results do not support OB and CO. Later on, you clarify this but this is not correct. Also, clarify how the ranking of predictors (PBC > EA > SN...) was derived—refer to ANN output explicitly.

Comment 11

The discussion should better contextualize results for art students specifically. References used (e.g., Hossain et al., 2021) may not reflect cultural or institutional realities in your sample. A local perspective would strengthen your interpretation.

Comment 12

Theoretical, practical implications and limitations should be integrated into the conclusion section for better flow and structure.

Comment 13

There are several minor English language issues throughout the manuscript. A final proofreading is strongly recommended.

Reviewer #3: Dear Author,

I am very pleased with the proposed manuscript that explores the factors that influence entrepreneurial intention among art university students, focusing on how the Diffusion of Innovation Theory and the Theory of Planned Behavior can explain these intentions. The manuscript highlights that perceived behavioral control is the strongest predictor of entrepreneurial intention, followed by entrepreneurial attitude and subjective norms. The study uses survey data and statistical analysis (SEM, ANN) to provide insights that can help universities and policymakers support entrepreneurship among art students. Overall, the article presents a sound and well-structured analysis of the factors influencing entrepreneurial intention among art university students. It is a good piece of work that effectively applies established theories to provide valuable insights for promoting entrepreneurship in this unique group.

6. PLOS authors have the option to publish the peer review history of their article (what does this mean? ). If published, this will include your full peer review and any attached files.

**Do you want your identity to be public for this peer review?** For information about this choice, including consent withdrawal, please see our Privacy Policy .

Reviewer #1: No

Reviewer #2: No

Reviewer #3: No

---

## [Author Response · Author response to Decision Letter 1]

10 Jul 2025

Editor's comments:

Comment 1:

The title should be shortened - I suggest omitting the second part of the title, since combining TPB and DOI in research of this type is not new

Response to Comment 1:

Thank you for your insightful suggestion regarding the title. We fully agree that the combination of TPB and DOI is no longer considered novel in this research domain and that a more concise title would improve readability. In response, we have revised the title to remove the theoretical model references and instead highlight the methodological contribution of our study. The new title reads:

Predicting Art University Students’ Entrepreneurial Intention: A Hybrid SEM–ANN Approach

We believe this revised title is shorter, clearer, and more aligned with the methodological focus of the manuscript, while maintaining the core subject matter.

Comment 2:

I suggest proofreading the paper to remove spelling and stylistic language errors

Response to Comment 2:

Thank you for your comment. We appreciate your suggestion and have thoroughly proofread the manuscript to correct all identified spelling, grammar, and stylistic issues. We have revised sentence structures where necessary to improve clarity, consistency, and academic tone. The entire manuscript has been edited using professional academic English standards to enhance readability and presentation quality.

Comment 3:

AMOS is the name of the software, not the data analysis method so that I would exclude it from the list of methods in the introduction

Response to Comment 3:

Thank you for pointing this out. We agree that AMOS is a software tool rather than a data analysis method. In response, we have revised the introduction to remove “AMOS” from the list of analytical methods. Instead, we now refer to the method as Structural Equation Modeling (SEM) to reflect the correct terminology. We appreciate your attention to detail, which has helped us improve the accuracy and professionalism of our manuscript.

Comment 4:

Figure 1 and Figure 2 should come with the source/reference since the models are not originally yours。

Response to Comment 4:

We have added the references for both theoretical models accordingly.

Comment 5:

It is necessary to provide insight into the items by which the variables were measured, along with references

Response to Comment 5:

Thank you for your valuable suggestion. We have revised the original list of measurement scales into a tabular format to enhance clarity and readability. A summary table has been added before Section 3.3, which includes three columns: scale name, source, and example item. This table has been incorporated into the latest version of the manuscript.

Comment 6:

What is shown in Table 5? It is not mentioned in the text at all

Response to Comment 6:

The original Table 5 has been renumbered as Table 6 in the revised manuscript. Table 6 presents the results of the VIF test. Moreover, Table 6 has been marked in the manuscript.

Comment 7:

The discussion should be more refined, enriched by a better elaboration of the results' similarities and differences with other comparable studies, and with more theoretical implications

Response to Comment 7:

Thank you very much for your insightful comment. We fully agree with your suggestion that the discussion should be better contextualized to reflect the specific cultural and institutional background of Chinese art university students. In response, we have substantially revised the relevant sections of the discussion to address this issue.

We enriched our interpretation of Perceived Behavioral Control by referencing Chinese art education practices such as project-based learning, national competitions (e.g., “Internet+ Innovation and Entrepreneurship Competition”), and creative incubation platforms, all of which are common in Chinese art universities and significantly shape students’ entrepreneurial self-efficacy.

For Entrepreneurial Attitude, we added a discussion of government-supported initiatives like the “Double Innovation” policy, and how interdisciplinary entrepreneurship education in Chinese art institutions nurtures a more favorable attitude toward creative entrepreneurship.

Regarding Subjective Norms, we removed the earlier reference to Hossain et al. (2021) and replaced it with a localized explanation grounded in the collectivist cultural context of China, emphasizing the influence of family, mentors, and peer networks in shaping entrepreneurial intentions among Chinese art students.

We also revised our discussion of Entrepreneurial Motivation and Relative Advantage by highlighting the rising importance of cultural and creative industries in China, and how art students’ intrinsic motivations (e.g., autonomy, cultural expression, personal branding) align with these industry trends.

Finally, for the non-significant results related to Compatibility and Observability, we proposed a culturally grounded interpretation: Chinese art students may prioritize self-expression and artistic values over external signals of entrepreneurial compatibility or visibility, which may weaken the effects of CO and OB in this specific population.

Reviewer 1

Comment 1:

The literature review is thorough but could benefit from more international comparative studies (For instance Pejic Bach, M. et al. (2018). Examining determinants of entrepreneurial intentions in Slovenia: applying the theory of planned behaviour and an innovative cognitive style. Economic research-Ekonomska istraživanja, 31(1), 1453-1471.; Hossain, M. I., Tabash, M. I., Siow, M. L., Ong, T. S., & Anagreh, S. (2023). Entrepreneurial intentions of Gen Z university students and entrepreneurial constraints in Bangladesh. Journal of innovation and entrepreneurship, 12(1), 12.

Response to Comment 1:

Thank you very much for your valuable and constructive feedback on our manuscript. We sincerely appreciate your comment regarding the need to include more international comparative studies in the literature review.

In response, we have revised the literature review section to incorporate comparative research findings from different cultural and national contexts. Specifically, we have added the following studies:

(1)Pejic Bach et al. (2018), who examined entrepreneurial intentions among Slovenian students using the Theory of Planned Behavior and innovative cognitive style, highlighting the moderating effects of cultural and cognitive dimensions.

(2) Hossain et al. (2023), who focused on Gen Z university students in Bangladesh and explored entrepreneurial constraints in a developing economy context.

These additions help broaden the theoretical scope and provide a richer comparative understanding of how entrepreneurial intention is shaped across different socio-economic and cultural settings.

Comment 2:

explain sample selection criteria

Response to Comment 2:

Thank you for your suggestion. We have added a detailed explanation of the sample selection criteria in the methodology section. Specifically, we clarify that participants were full-time art university students who had completed at least one year of study and voluntarily agreed to participate. We also describe the recruitment channels and rationale for choosing this population to ensure relevance to the study's objectives.

Comment 3:

The use of convenience sampling from only three universities may limit generalizability. A discussion of sampling limitations should be more prominent in the Discussion section.

Response to Comment 3:

Thank you for highlighting this important point. In the revised Discussion section, we have expanded the limitations subsection to more clearly acknowledge the constraints posed by the use of convenience sampling from only three institutions. We now explicitly discuss how this may impact the generalizability of the findings and recommend that future research adopt more diverse and representative sampling strategies.

Comment 4:

Common method bias is not address. Adding a procedural or statistical would strengthen the validity.

Response to Comment 4:

Thank you very much for your valuable suggestion regarding common method bias. We fully agree that addressing this issue is essential for enhancing the validity of the research findings.

We conducted Harman's single-factor test, and the results showed that the first factor accounted for only 32.4% of the total variance, which is well below the critical threshold of 50%. This indicates that common method bias is not a serious concern in this study.

The above information has been added to the revised manuscript in Section 4.1 Measurement Model.

Comment 5:

While SEM-PLS is valid, the rationale for using this method over CB-SEM could be briefly elaborated.

Response to Comment 5:

Thank you for your valuable comment regarding the use of SEM-PLS. We agree that clarifying the rationale for choosing this method over CB-SEM is important for methodological transparency.

In response, we have added a detailed explanation in the Data Analysis section of the revised manuscript. Specifically, we chose PLS-SEM over CB-SEM for the following reasons:

(1)PLS-SEM is more appropriate for exploratory research and predictive modeling, which aligns with our study’s objective of identifying key predictors of entrepreneurial intention among art students.

(2) PLS-SEM handles complex models with multiple constructs more flexibly, especially when combining theoretical frameworks (DOI and TPB).

(3) PLS-SEM has fewer assumptions regarding data distribution and sample size. Given our sample of 273 and the lack of multivariate normality, PLS-SEM provides more robust and accurate results in this context.

Comment 6:

For those of us not familiar with provinces in China, please state in your paper that Liaoning Province is in China

Response to Comment 6:

Thank you for your helpful suggestion regarding geographic clarity. In response, we have revised the manuscript to explicitly state that Liaoning Province is located in China. This clarification has been added to both the Abstract and the Methods section to ensure accessibility for an international audience.

Comment 7:

The discussion restates findings well but could be enriched by more theoretical implications, particularly how EI in creative fields differs from technical/business contexts.

Response to Comment 7:

Thank you very much for your thoughtful comment regarding the theoretical implications. We agree that distinguishing the nature of entrepreneurial intention in creative versus technical/business contexts is essential for enhancing the academic contribution of this study.

In response, we have revised Section 5.1 Theoretical Implications to elaborate on how EI in creative fields is more identity-driven, intrinsically motivated, and aesthetically oriented, in contrast to the more rational and profit-oriented nature of EI in technical or business domains. This addition aims to expand current theoretical frameworks by emphasizing domain-specific differences in entrepreneurial intention formation.

Comment 8:

Consider reflecting on how cultural factors in China may shape EI in ways that might not generalize elsewhere.

Response to Comment 8:

Thank you for your helpful suggestion on considering cultural influences. We have added a brief reflection in the Limitations and Future Directions section, noting that cultural factors such as collectivism and Confucian values may shape entrepreneurial intention differently in China compared to other contexts. We also suggest that future studies explore this issue through cross-cultural comparisons.

Comment 9:

The language is generally clear, though some minor grammatical issues and awkward phrasing appear.

Response to Comment 9:

Thank you for your suggestion. We have carefully proofread the entire manuscript and revised several sentences to improve clarity, grammar, and overall readability. We have also ensured consistency in terminology and corrected minor language issues throughout the text.

Reviewer 2

Comment 1:

In sections 2.1 and 2.2, use full names ("Diffusion of Innovation Theory" and "Theory of Planned Behavior") in titles instead of acronyms.

Response to Comment 1:

Thank you very much for your valuable comments and suggestions, which have helped us further improve the quality and clarity of the manuscript. Based on your feedback, we have made the following revisions:

(1) The original section title “2.1 DOI” has been revised to “2.1 Diffusion of Innovation Theory”;

(2) The original section title “2.2 TPB” has been revised to “2.2 Theory of Planned Behavior”.

These changes enhance the clarity and readability of the section headings and make the manuscript more accessible to a broader academic audience.

Comment 2:

The explanation of selected variables in 2.1 should be moved to the methods section. Also, please clarify why certain DOI variables were excluded. If complexity is the issue, specify what kind—statistical, theoretical, or interpretative?

Response to Comment 2:

Thank you very much for your valuable comments. In response to your suggestions, we have made the following revisions:

1.We have relocated the explanation regarding the selection of DOI variables (i.e., why we chose “Relative Advantage,” “Compatibility,” and “Observability”) from Section 2.1 to the methodology section. A new subsection titled “3.4 Variable Selection Rationale” has been added to clearly distinguish between theoretical background and research design.

2.Regarding the exclusion of other DOI dimensions such as “Complexity” and “Trialability,” we have provided a detailed explanation in the methodology section. Specifically, the exclusion was based on considerations of both statistical and interpretive complexity:

(1) From a statistical perspective, including all five DOI dimensions would have significantly increased the number of model paths and computational burden, potentially compromising model parsimony and stability—especially given our moderate sample size (n = 273).

(2) From an interpretive standpoint, prior studies (e.g., Venkatesh et al., 2012) have pointed out that in entrepreneurship research—particularly in creative industry contexts—the effects of “Complexity” and “Trialability” are often unstable or unclear. Therefore, we prioritized three core dimensions that have demonstrated stronger empirical validity and more consistent measurement in previous research.

Comment 3:

In the TPB section of literature review, specify which universities or types of art schools the students come from. Perhaps discussion of the results from the literature regarding differences in TPB when different students are considered (business school, medicine, law...) Also, consider expanding the TPB discussion from the entrepreneurial intention perspective, as there is rich literature in this domain.

Response to Comment 3:

Thank you for your valuable suggestion. In response, we have added a comparative discussion in the TPB section of the literature review to address how TPB constructs may vary across different academic disciplines. Specifically, we noted that business students tend to exhibit higher perceived behavioral control due to structured entrepreneurship education, whereas students in disciplines such as medicine or law often demonstrate lower entrepreneurial attitudes because of fixed or institutionally regulated career pathways.

We also highlighted the distinctive characteristics of art university students. These students are typically driven by intrinsic creativity and personal expression, which may enhance their entrepreneurial attitude but also make them more sensitive to social expectations. Furthermore, in collectivist cultures like China, family and institutional pressures often play a strong role in shaping subjective norms. By incorporating these insights, the study more accurately reflects the contextual and disciplinary dynamics that influence entrepreneurial intention among art university students.

We appreciate this suggestion, which has helped us better situate our study within the broader TPB literature and emphasize its relevance to student populations with diverse academic backgrounds.

Comment 4:

The integration of DOI and TPB is conceptually promising. However, before 2.3, clarify how the two theories are

---

## [Editor Report · Decision Letter 1]

30 Jul 2025

PONE-D-24-60132R1Predicting Art University Students’ Entrepreneurial Intention: A Hybrid SEM–ANN ApproachPLOS ONE

Dear Dr. Zhang,

Thank you for submitting your manuscript to PLOS ONE. After careful consideration, we feel that it has merit but does not fully meet PLOS ONE’s publication criteria as it currently stands. Therefore, we invite you to submit a revised version of the manuscript that addresses the points raised during the review process. =====================

The language and formal organization of the document are not completely edited and aligned. For example:1. .*..factor ’ s relative importance.* - part of the abstract, unnecessary space before *s *2. Between the text and the parentheses in which the reference or abbreviation is listed, a space is required3. *The structure of this paper is as follows: The second section presents the literature review and research hypotheses.* After the word *follows* a dot is needed, not a colon.4. *The TPB has five key constructs: attitude, SN, perceived133 behavioral control (PBC), behavioral intention, and actual use (Fig. 2). * Why is SN in the abbreviation, and the rest is not? The style should be consistent. These are just some examples. The entire text should be edited. Keep in mind that language and writing style are one of the PLOS ONE criteria for publication (PLOS ONE does not copyedit accepted manuscripts).

PLOS ONE criteria:

T*he article is presented in an intelligible fashion and is written in standard English.*

*PLOS ONE does not copyedit accepted manuscripts, so the language in submitted articles must be clear, correct, and unambiguous. We may reject papers that do not meet these standards.*

*If the language of a paper is difficult to understand or includes many errors, we may recommend that authors seek independent editorial help before submitting a revision. These services can be found on the web using search terms like "scientific editing service" or "manuscript editing service."  *
https://journals.plos.org/plosone/s/criteria-for-publication#loc-5

We look forward to receiving your revised manuscript.

Kind regards,

Tamara Šmaguc, Ph.D

Academic Editor

PLOS ONE
---

## [Author Response · Author response to Decision Letter 2]

1 Aug 2025

Response to Reviewer’s Comment

Reviewer Comment 1: “…factor ’ s relative importance. - part of the abstract, unnecessary space before s.”

Response: Thank you for pointing out this typographical issue in the abstract. We have carefully corrected the spacing error by removing the unnecessary space before the apostrophe ’s in “…factor’s relative importance.” In addition, we conducted a meticulous line-by-line proofreading of the entire manuscript to identify and correct any other spacing or punctuation inconsistencies. This ensures alignment with the PLOS ONE criteria regarding clear, correct, and unambiguous English.

Furthermore, as PLOS ONE emphasizes the importance of professional language presentation, we have revised the manuscript thoroughly for standard academic English style and clarity. Where necessary, we also consulted professional editorial support to enhance the overall language quality.

We sincerely appreciate this helpful observation.

Reviewer Comment 2: “Between the text and the parentheses in which the reference or abbreviation is listed, a space is required.”

Response: Thank you for bringing this formatting issue to our attention. In accordance with your suggestion and the formatting standards required by PLOS ONE, we have carefully reviewed the entire manuscript and inserted a space between the preceding text and all parentheses containing references or abbreviations (e.g., “…as suggested by previous studies (Smith et al., 2022)” → “…as suggested by previous studies (Smith et al., 2022)”).

We have ensured that this spacing adjustment has been applied consistently throughout the manuscript to maintain professional presentation and alignment with the journal’s formatting expectations.

Reviewer Comment 3: “The structure of this paper is as follows: The second section presents the literature review and research hypotheses. After the word follows a dot is needed, not a colon.”

Response: Thank you for identifying this punctuation inconsistency. In accordance with your suggestion, we have revised the sentence by replacing the colon with a period after the phrase “The structure of this paper is as follows.” The corrected sentence now reads:

“The structure of this paper is as follows. The second section presents the literature review and research hypotheses…”

We have also carefully reviewed the manuscript for similar punctuation issues and have ensured consistent and proper use of colons and periods throughout. Your attention to detail is greatly appreciated and has helped us improve the clarity and formal presentation of the paper.

Reviewer Comment 4:“The TPB has five key constructs: attitude, SN, perceived behavioral control (PBC), behavioral intention, and actual use (Fig. 2). Why is SN in the abbreviation, and the rest is not? The style should be consistent.”

Response: Thank you for your thoughtful comment regarding the consistency of abbreviations in our description of the Theory of Planned Behavior (TPB) constructs. We would like to clarify that, in this instance, we intentionally did not abbreviate “attitude” and “behavioral intention” because these constructs are not directly applied in our study and are introduced only as part of the theoretical background. In contrast, both “subjective norm (SN)” and “perceived behavioral control (PBC)” are used not only within the TPB framework but also directly adopted in our research model, where their abbreviated forms are consistently applied.

Moreover, “SN” was previously introduced and abbreviated in an earlier section, which is why it appears in abbreviated form here. We have ensured that the usage remains consistent throughout the manuscript based on whether the construct is contextually relevant to our model or mentioned only in the theoretical overview.

We appreciate your attention to this detail and hope this explanation clarifies our rationale for the selective use of abbreviations.

---

## [Editor Report · Decision Letter 2]

7 Aug 2025

Predicting Art University Students’ Entrepreneurial Intention: A Hybrid SEM–ANN Approach

PONE-D-24-60132R2

Dear Dr. Zhang,

We’re pleased to inform you that your manuscript has been judged scientifically suitable for publication and will be formally accepted for publication once it meets all outstanding technical requirements.

Kind regards,

Tamara Šmaguc, Ph.D

Academic Editor

PLOS ONE
---

## [Editor Report · Acceptance letter]

PONE-D-24-60132R2

PLOS ONE

Dear Dr. Zhang,

I'm pleased to inform you that your manuscript has been deemed suitable for publication in PLOS ONE. Congratulations! Your manuscript is now being handed over to our production team.

Kind regards,

on behalf of

Asst.Prof. Tamara Šmaguc

Academic Editor

PLOS ONE